# Health Providers’ Response to Female Adolescent Survivors of Sexual and Gender-Based Violence and Demand Side Barriers in the Utilization of Support Services in Urban Low-Income Communities of Nigeria

**DOI:** 10.3390/healthcare11192627

**Published:** 2023-09-26

**Authors:** Olutoyin Opeyemi Ikuteyijo, Andrea Kaiser-Grolimund, Michael D. Fetters, Akanni Ibukun Akinyemi, Sonja Merten

**Affiliations:** 1Swiss Tropical and Public Health Institute, 4123 Allschwil, Switzerland; andrea.kaiser-grolimund@swisstph.ch (A.K.-G.); sonja.merten@swisstph.ch (S.M.); 2University of Basel, 4001 Basel, Switzerland; 3Mixed Methods Program, Faculty of Medicine, University of Michigan Ann Arbor, Ann Arbor, MI 48104, USA; 4Department of Demography and Social Statistics, Obafemi Awolowo University Ile Ife, Ife 220005, Nigeria; akakanni@oauife.edu.ng

**Keywords:** health providers, female adolescents, sexual and gender-based violence, barriers, low-income communities

## Abstract

Survivors of sexual and gender-based violence (SGBV) are often hindered in their quest to access quality healthcare. This has a significant effect towards the achievement of Sustainable Development Goal SDG Target 3.7. to ensure universal access to sexual and reproductive healthcare services. This study is focused on identifying some of the demand side barriers in accessing health care services, particularly among young girls who are survivors of SGBV within intimate relationships in poor urban areas in Nigeria. The study used an ethnographic approach to solicit information from health providers, adolescents, and young women (AYW) in 10 low-income communities in two major cities in Nigeria, Ibadan and Lagos. Findings showed that there are structural limitations within the primary health care (PHC) system that posed a great challenge for survivors of SGBV to access services. Some of these include non-existing counseling services, a lack of rehabilitation centers, poor referral, and a lack of training for health providers in handling survivors of SGBV. There is also a lack of skills among health service providers that have negative influence on support services to survivors of SGBV. On the demand side, poor knowledge of possible health seeking pathways, a lack of education, and social support are barriers to accessing appropriate services among adolescent and young SGBV survivors. The study concluded that integrated services at the PHC level should include adequate and timely treatment for survivors of SGBV and targeted intervention to upscale skills and knowledge of health care providers.

## 1. Background

Around the world, female adolescents and young women are at risk of sexual and gender-based violence perpetuated by their intimate male partners. Globally, statistics have shown that nearly one in three adolescent girls (aged 15–19 years), or about 84 million, have been victims of physical, emotional, and sexual violence perpetrated by their husbands or intimate partners. Additionally, about 15 million have experienced forced sex, and one in five women aged 20–24 were married before 18 [1].

The well-being of young people, particularly young women is a critical developmental agenda and featured prominently in the Sustainable Development Goals (SDG) [2]. According to the Secretary General of the United Nations, “Adolescents are central to everything we want to achieve, and to the overall success of the 2030 Agenda” [3]. Within the African continent, The Maputo Plan of Action (2016–2030), and The African Agenda 2063 all converged on the need to prioritize investment in ensuring a healthy and productive young generation, as the future of adolescents is essential in the development of societies.

The age of adolescence relates to diverse experiences that are, however, different from experiences of other age groups, which explains the need for specific programs and interventions targeting the diverse group of adolescents [4]. Initiatives such as the Global Strategy for Women’s Children’s and Adolescents’ Health 2016–2030 and the WHO Global Accelerated Action for the Health of Adolescents (AA-HA!) [4] have called for research on adolescents’ sexual and reproductive health as well as their experiences of gender-based violence. Research has shown that in Sub-Saharan Africa, 33% of women aged 15–49 years have experienced SGBV in their lifetime, and 20% experienced SGBV in the previous year [5]. Additionally, 1 in 4 adolescent girls who have been in a relationship had experienced all forms of SGBV [5].

Nigeria is a signatory and has made commitment to several international developmental agendas and initiatives, including the SDG [2], the International Conference on Population and Development (ICPD) and ICPD Beyond 2014 Follow-Up Action [6], and the Convention on Elimination of All Forms of Discrimination against Women (CEDAW) [7]. Despite Nigeria’s commitment, nearly 3 in 10 women have experienced physical violence by the age of 15 years [8], and 9% had experienced sexual violence in the previous year [9]. At sub-national level, a study from Calabar, in southeastern Nigeria among teenage girls, found that beatings by sexual partners and guardians were prevalent in the study area [10] while a study from the eastern part of Nigeria also found a high prevalence of violence among adolescents, with parents and relatives being the major perpetrators [11]; in Lagos a study among adolescents found that 36% initiated sex by coercion and 64% believed sexual violence was common in the community [12].

Nigeria is a multi-cultural and multi-ethnic society with over 500 languages and about 250 ethnic groups. However, across the country, a patriarchal system dominates Nigerian societies providing an environment for SGBV. Within this context of gender inequity and cultural norms, the magnitude of some of these gender-based vices continue to increase. Although the government has enacted the Violence Against Peoples Prohibition Act [13], this has not translated into the desired outcome [14]. Despite the high incidence and magnitude of SGBV, there is also a lot of concern for care and support for survivors, particularly for their health and general well-being.

AYW face countless barriers in utilizing services for their health and well-being, particularly the survivors of GBV. The demand side of services by AYW, ranging from psycho-social support, counseling, justice, and other therapies, are influenced by many factors. The first step is the ability to report to the appropriate service. However, this is overshadowed by some values and restraints which are described as “culture of silence” [15] a social norm that is often referred to as common among women experiencing SGBV in an intimate relationship “The culture of silence is described as the behavior of a group of people that by unuttered agreement do not mention, discuss, or acknowledge a particular subject. In this context, it is the act of keeping an unspoken agreement not to speak about what happened to them”survivor [15]. Not only in Nigeria, but also in other contexts, adolescents and young women find it difficult to speak out due to shame, stigma, and some socio-culturally shaped ideation [16] about retaliation effects on the survivors [17,18]. In addition, depending on the relation to the state, there is also a perceived notion of a lack of confidence in the state agencies, including the healthcare systems [19,20]. While there has been an increase in youth-friendly sexual and reproductive health services (YF-SRHS) that target contraceptive uptake of young people in Nigeria, there is still a gap in integrating SGBV services into this youth friendly initiative.

An extension of youth-friendly services to issues of violence among young people will enhance disclosure of SGBV in intimate relationships. There is evidence that family planning clinics that provided education about gender-based violence improved health providers’ knowledge of handling SGBV cases and support to survivors of SGBV [21]. However, in the past, the support services available to adolescents and young women in Nigeria have been mostly informal, i.e., families, friends, and religious bodies [22]. This support has not necessarily translated into the desired improvement for mitigating SGBV experiences among young women. Hence, there is a need to focus on improving the health system’s preparedness to address and support the survivors of SGBV, particularly vulnerable adolescent girls.

### 1.1. The Role of Health Providers

Healthcare services are essential in mitigating the consequences of violence among women, especially the most vulnerable group of female adolescents and young women living in low-income neighborhoods or slums. Globally, healthcare facilities remain one of the key entry points for survivors of violence among AYW to provide diverse support services [23] as to seek treatment for sexual, physical, psychological, and other reproductive morbidity due to a violent experience. The World Health Organization (WHO) manual [24] identified healthcare providers as critical in addressing SGBV. According to the manual, they are an important agency in “identifying a survivor, facilitating access to support services, contributing to preventing the recurrence of violence, integrating into health education and health promotion with clients, involvement in community awareness about human rights and documenting the magnitude of the problem for advocacy”. A Lancet publication on violence against women identified the healthcare facility as a safe environment for survivors of violence that will aid in open disclosure and appropriate support systems for survivors [25].

There is a dearth of studies that addressed specific sectors of the health system (psychology, psychiatry, counseling, and rehabilitation centers) on their roles in combating violence in Nigeria. Healthcare providers are critical stakeholders in improving the health outcomes of SGBV survivors [26]. Studies on SGBV from Nigeria have largely focused on issues of justice and the involvement of legal authorities, such as the police [27,28,29]. Other studies available are mostly from high-income countries [20,21,30].

### 1.2. Intersectionality in Healthcare Service for GBV Survivors

Intersectionality theory is a systemic perspective [31] to understand how systems of power and privilege constitute people’s experiences of oppression and marginalization. It addresses various ways in which individuals belong to multiple social positions simultaneously such as age, gender, ethnicity, cultural background, socioeconomic class, sexual orientation, religious belief, dis/ability status, and citizenship status, and how this shapes people to continuously inform positions of social power relations [32,33].

From a health system perspective, intersectionality brings attention to diversity within population groups that were considered homogenous, giving explanations to nuances surrounding vulnerabilities and the intricate nature of health inequities [34]. Intersectionality is regarded as an approach to fostering a multifaceted analysis of power structures and relations that interact to produce and sustain inequalities in various health outcomes [31,34,35] by addressing the power dynamics, which invariably exist at the center of violence experienced by AYW. Intersectional violence occurs in terms of place and time “The term “survivor” in this text refers to adolescents and young women who have experienced sexual and gender-based violence at some point in their intimate relationship and who have suffered pain and injury but not death. The term “survivors” also refers to those women who are currently still in an abusive relationship” as systemic domination and exclusion to form a unique oppression that is typically not detected by traditional methods and modes of interpretation [33]. This focus on power dynamics first encompasses how different experiences of violence create diversity in terms of support (social and health), what is available, and social locations of privilege and disadvantage. City dwellers in marginalized areas are exposed to poor healthcare services due to low socioeconomic status and environmental conditions. AYW are disadvantaged due to the location where they reside and linked the lack of available services. The majority of health professional services (psychologists, psychiatrists, counselors, and rehabilitation centers) that would give support in the process of recovery from violence are not available in most or any of the PHC facilities in the low-income urban communities in Nigeria. However, these services are essential to mitigate experiences of SGBV among AYW and also to prevent re-victimization and morbidity among this group. Increased risk of SGBV among female AYW creates differential burdens with less access to formal support services. Each of these services is important although they were less emphasized in the past, especially in the Nigerian context where these services were not sufficient or ever existed and were most needed.

Given the lack of research about health system support towards violence against AYW, the purpose of this study was to look at the perceived barriers encountered by AYW (solely on violence experienced within relationships) in accessing health services, and the health providers’ responses and challenges faced in providing adequate services to survivors of GBV in two low-income areas of southwest Nigeria.

## 2. Methods

### 2.1. Study Design

The study used different qualitative research methods that were triangulated for cross-validation of research findings. A combination of ethnographic methods, specific observations, informal conversations, key informant interviews (KII), in-depth interviews (IDI), and focus group discussions (FGD) were used in eliciting information from two locations in two Nigerian cities. We conducted KII with 10 health providers, IDI with 40 adolescent girls and young women, and 9 sessions of FGD with a 166 minimum of 8 participants in each group from January 2021 to June 2021 in both locations. The data used in this study form part of a larger ethnographic study. An ethnographic approach was employed to gain a deeper understanding of the urban context as well as to better understand the pathways of care and the daily life experiences of people who lived in the different selected communities of the study. To this end, the research team had to visit the communities and identify key stakeholders who served as gatekeepers for the study. The research team also lived in the communities for the period of data collection. Some particular spots and facilities were identified by the first author in the process of a transect walk through the communities. These facilities and spots included healthcare facilities (private and public) usually patronized by survivors of SGBV and the nature of living conditions (washroom outside the building) that could make AYW vulnerable to SGBV. The dark spots and highly dangerous areas (incomplete buildings) identified by community members where some acts of SGBV have been perpetrated in the past were among other observations. Additionally, follow-up interviews were conducted with survivors who attended health facilities for further and in-depth discussions. The selection of the health facilities was purposive as these facilities were situated within each of the communities of the study locations. The research team purposively interviewed health providers who were trained in youth-friendly sexual and reproductive health services (YF-SRHS) in the PHC facilities in the communities, as they were closer to AYW and would have provider perspectives on determinants of access. The study hoped thereby to learn from their perceptions and experiences of the specific challenges and relative disadvantages faced by both survivors and providers in each community.

### 2.2. Study Instruments

A semi-structured interview guide was utilized and was verified by experts from medical sociology and epidemiology. Participants were invited to share their experiences of violence with their previous or current partners, their stories of different intimate partner violence in their relationships, and barriers in reporting to health providers in seeking help or support. Health providers were asked to share their knowledge about various interventions/support given to survivors of SGBV, the challenges they faced in providing adequate support to the survivors, and their perception on the inadequacy of the health system to give the support necessary to AYW survivors. Interviews were complemented by the observations from health providers at primary healthcare centers by an analysis of center and facility protocols for violence.

### 2.3. Study Setting and Recruitment Procedure

The study was conducted in the informal settings of two major metropolitan cities of Lagos and Ibadan in Nigeria from January to June 2021. Lagos is the biggest metropolitan city with a population projected at 9 million, and Ibadan’s population was projected at 3,565,108, making it the third largest metropolitan city as of 2022 [36]. They are the major cities in the Southwest of Nigeria with the largest low-income settlements. In 2022, the low-income population in Nigeria was estimated to be 55% of the total population [37], the local government areas (LGA) in both locations of the study were identified as major low-income areas in Southwest Nigeria. The study locations are characterized as having poor social and health infrastructures. The first author spent time in the two sites before the study to familiarize herself with the location and to gain trust of the residents. The AYW for FGD sessions were recruited through health providers and community leaders. Due to the sensitive nature of the topic for the interviews, most of the survivors who had experienced any form of violence in the past were recruited using a snowball method; others were also referred to the study team through the health providers. Snowball sampling is a non-probability sampling method where new units are recruited by other units to form part of the sample. This method is used when working with a population that is difficult to access, such as the participants of this study (survivors of SGBV). After the first respondent was recruited, she helped to connect to her friend who had experienced SGBV; the link continued till saturation was reached. The study team informed them about the study and asked them whether they were interested in sharing their experiences before the interview.

### 2.4. Study Scope

The study scope was limited to female AYW who resided in the two study locations and to health providers who worked in these study areas. The study was conducted in the selected low-income communities of Agege in Lagos and Ibadan Southeast LGA in Ibadan.

### 2.5. Study Population and Data Collection Procedure

The population of the study included AYW and health providers in the study locations. The participants were female adolescents and young women living in the Ibadan Southeast and Agege and Orile-Agege LGAs in Lagos. In Lagos, Orile-Agege is an annex of Agege LGAs, the communities under both are inter-connected as a result of political demarcation. The selection included female adolescents and young women who were residing in the communities for at least twelve consecutive months and who had experienced violence in the past years in intimate relationships. The selection criteria for health providers was their work in YF-SRHS in the PHC facilities situated in both sites in Lagos and Ibadan. The in-depth and KII lasted approximately 45 min and were audiotaped in the open data kit (ODK) platform to guide against loss of interview information and were sent to the server immediately. The participants were duly informed about the purpose of the study and consent forms were signed, although some participants’ preferred to provide their verbal consent because they wanted to protect their identities. They were informed that if they were not comfortable, they could have the possibility to stop the interview at any time. Depending on their preferences, private places in their homes and also quiet spaces at PHC centers were used in order to establish a trustful atmosphere for the interviews. One of the survivors preferred a phone interview with follow-up at later stage. Incentives in the form of toiletries were provided for the AYW participants while a moderate amount of call cards were provided for the health providers. Two female research assistants working with local ‘non-governmental organization (NGO) with Masters’ degrees in Sociology and Demography, respectively, and with experience in qualitative research conducted the interviews alongside the first author.

### 2.6. Data Analysis

The objectives of the study guided the analysis of the data. The ODK was used for the collection of information through interviews, and data were transcribed verbatim. The Atlas ti version 8, a computer-assisted qualitative data analysis software (CAQDAS) was used in sorting the transcripts into codes and themes. An inductive approach through thematic content analysis was developed through in-depth reading of the transcripts and iterations; codes and themes were developed. The main themes were based on the interview guide, but the sub-themes emerged directly from the participants’ discussions. Additionally, we used framework analysis [38] to extract factors responsible for not reporting abuse or utilizing health services for support and health providers’ responses. The steps recommended by Gale and colleagues were followed: (1) transcription; (2) familiarization with the interview; (3) coding; (4) developing a framework; (5) applying the framework; (6) charting data into a framework matrix; and (7) interpreting the data [38]. These steps guided the analysis with the inclusion of the observational data, informal discussions with respondents, daily debriefing of the research team, field notes, and reflections after each interview. Reporting was guided by Consolidated Criteria for Reporting Qualitative Research (COREQ-32) [39]. Four categories emerged as follows: (1) factors responsible for not reporting abuse or utilizing health services for support; (2) commonly known support system available for AYW in the event of violence; (3) perceived barriers in accessing services or help by abused AYW; and (4) health providers’ responses, challenges, and involvement on AYW recovery process.

## 3. Results

The sociodemographic characteristics of the participants in this study are shown below.

The two locations of the study were Lagos and Ibadan low-income communities. Twenty female AYW were interviewed one-on-one in each location. Only four sessions of FGD were conducted in four communities in Ibadan, and five were conducted in five communities in Lagos. The age range of the female adolescent participants was (15–24), more than three-quarters of the participants were below 19 years, while the rest were above 20 years. The majority of the adolescents were apprentices while the rest were students, with the highest education having secondary school certificates. Greater than an average was married (cohabiting) while the rest were single and/or living alone. More than the average of the participants reported physical abuse while the other participants reported sexual and economic abuse. There were five healthcare providers from each location, making a total of ten. More than the average number of the healthcare providers were female, either doctors, matrons, or health counselors. Each of the participants from the healthcare providers’ sides was trained in sexual and reproductive health but very few explicitly in gender-based violence response to victims. The majority of the healthcare providers had well over twenty years of experience in their professional field.

The results were organized from two perspectives: (1) from the AYW level, we examined experiences of violence and barriers to reporting, and (2) from the healthcare providers’ levels, we looked at providers’ responses and challenges in treating GBV survivors in primary healthcare centers.

### 3.1. Violent Experiences and Barriers in Reporting: Survivors’ and Health Providers’ Perspectives

#### 3.1.1. Factors Responsible for Not Reporting Abuse or Utilizing Health Services for Support

##### “Culture of Silence” among Survivors of Abuse

As described in the literature, given the sensitive nature of sexual violence, especially in low-income communities, a culture of silence prevails, forcing the survivor to remain silent about the incident of violence. There are several intersecting layers of inequality inherent in such a culture of silence, including male dominance, where men can abuse women while blaming and threatening the survivors to evade any consequences. It is also a culture where the rich can abuse the poor who are not (sufficiently) protected by the state or legal system, reflecting the inequalities that exist in urban space. In addition, healthcare providers participating in this study confirmed this prevalent norm in the study communities. It was that no one should know about the sexual status of survivors who found themselves in an abusive situation and survivors who had experienced SGBV. This is usually done to protect the identity of the survivor and, in some cases, also of the perpetrators if they are close relatives or if the perpetrator is a personality with power and influence in the community. Likewise, the tendency of married young women was to conceal their situation because they are trying to protect their homes and project a perfect image of a marriage, which has also exacerbated the culture of silence. A medical doctor in Sango, Lagos, explained that many women did not report abuses against them because of the fear of reprisals for reporting. They explained that:


*…It’s very rare cases where people come and report, so even there’s no forensic evidence and the fact that the community they live in when you report abuse or rape, how are other people going to perceive you? I think it is a cultural issue for the community, the lack of voices.*

*(Medical doctor, Sango Lagos)*


In addition, another health worker in Dopemu Lagos reported that it was against the communally shared cultural practices to report abuse because the young women (survivors) would be the laughing stock eventually if any report was made. This is an issue of stigmatization, which makes the majority of survivors not report abuse experiences. They elaborated:


*[…] Culturally, it is not so much acceptable, they will believe that “let us just cover it” so that at the end of the day, they will not be abusing the girl. Socially, they may be embarrassed so they will want to keep it […].*

*(KII Health worker, Dopemu Lagos)*


There are other agencies to which SGBV acts can be reported apart from health system, such as police and human rights organizations. However, most survivors did not make use of them, possibly because of a lack of awareness, financial constraints, or lack of confidence in the kind of support they might get from these agencies. As mentioned in an IDI:


*Some of such girls may report to the police, or human rights but majority will not report and keep it to themselves…*

*(IDI FA Dopemu Lagos)*


#### Stigma: A Situation of Being Judged Usually Discourages Reporting of SGBV

Closely linked to the aspect of silence mentioned above, the fear of stigmatization was also brought forward as another major discouragement for reporting any abuse by survivors. The term “survivors” also are those who have experienced any forms of violence and were able to escape, especially when it had not lead to death. Health workers in Lagos and Ibadan reported that some parents would even discourage their children, who had been survivors of abuse, from reporting. They feared the reactions of neighbors who might begin to spread the news of the abuse to the utmost embarrassment of the survivors. As a health worker in Lagos elaborated:


*[…] Most of the concern they face is parents discouraging them in reporting, then fear of stigma, fear of being shouted at too, fear of—I don’t know what—people around when they broadcast this issue now, I don’t know what my friends will see me as […].*

*(KII Health worker, Powerline Lagos)*


Some other concerns are blame by the family members or community people, on not taking the right decisions before going into violent relationships. Due to this pre-assumption, AYW usually stayed in the abusive relationship, as described in a FGD:


*People will blame them that before they started the relationship, why don’t they seek counsel, so if they will have to report. […]’ ‘[…] Because they are scared of what people are going to say, they are scared of being stigmatized so they stay in the abusive relationship’.*

*(FGD Dopemu, Lagos)*


##### Fear of Re-Victimization from the Perpetrator and Lack of Confidence in the Justice System

Another reason for not reporting IPV is the re-victimization of survivors by perpetrators of violence. Since some AYW are young and those who are married and have children still want to retain their marriage, the fear of separation and the responsibility of having to take care of the children alone keep them from reporting violence. An interview also revealed experience with the judicial system that allowed perpetrators to come back to the community “as if nothing happened”:


*[…] the perpetrators ran away to another environment immediately after they commit the crime, they returned to the community with the hope that people would have forgotten the incident. I witnessed a case where a girl was raped of recent the perpetrators were caught and sent to prison, they have been released now and back to the community, as if nothing happened.*

*(IDI AYW, Soretire Lagos)*


(1).Barriers in accessing support services and help for survivors

The majority of participants cited several barriers to reporting SGBV and accessing primary health care facilities in their localities, including blame, age, distance, threat from perpetrators, stigma, social acceptance of SGBV, shyness, religious belief, lack of resources, lack of awareness, delay, inaccessibility, children, and bonds of love. Some of these barriers are discussed below:

##### Socio-Economic Constraints of AYW and Their Families

Finances were brought forward as one of the major barriers to accessing health services in the communities studied. Given the relatively low socio-economic background of most inhabitants in the study locations, most of the participants reported that the cost of services served as a major barrier to accessing adequate support services, as this statement for a focus group discussion participant reveals:


*[…] I think that there is no way one will go to the hospital and they will not request for money, so if the person is capable and has the money she will go there.*

*(R8…FGD Oniyarin Ibadan)*


Hence, participants emphasized that whether victims decided to use the medical facilities or the services of the police, none were available for free as they all cost some money. AYW are either trainees, in school, or out of school and young, most of whom depend on their partners or parents for their livelihood. This limits their possibilities for accessing support, especially in terms of health services due to structural constraints, as this excerpt from a focus discussion reveals:


*[…] most of them go to the pharmacist and go to the hospital. If they have money, they will visit the hospital and the one that does not use the hospital finds an alternative.*

*(R2…FGD Oniyarin Ibadan)*


In addition, when it comes to reporting incidences at the police, the age of AYW and the financial aspect were brought forward, and the police perspective of SGBV hindered their service delivery:


*…the police cannot render any help, there is no police you go to that you will not spend money so there is no help police can render when it is not that we took a matter there, if she need support from police they cannot help her.*

*(R 5 FGD, Ogunpa Ibadan)*


See Figure 1.

##### Preferred Type of Services by Adolescents and Young Women in the Community

Participants were asked about the type of services they preferred, and most of the participants indicated that they preferred to go to a private hospital if they had the money to do so, as a result of the convenience and promptness of the services. However, there was a consensus of opinion that most young women would prefer to talk to private nurses in the case of rape. This shows the extent to which they would want to ensure privacy and avoid the use of hospitals as a result of stigmatization.


*[…] When it comes to rape, they go to private hospitals but not for beating. However, if is forced sex from their sexual partner, some people have nurses they confide in who comes to treat them at home.*

*(Resp. 4, FGD AYW Sango Lagos)*


The majority of AYW survivors still prefer private clinics for any experience of abuse compared to government owned facilities, because of privacy. This was mentioned by one of the participants:


*When it comes to pregnancy they go to the hospitals, it could be some government, or private depending on how much they can afford, but for beatings it usually chemist, and rape is definitely private or a nurse.*

*(IDI AYW, Dopemu Lagos)*


(2).Support systems available for AYW in the event of violence

In two of the ten communities in the study, there were other support services that had been organized by community groups and health care providers. According to the participants, these services included community networks to protect young women and other vulnerable groups against abuse. Interestingly, these community networks were well known by members of the community, and in some cases, they comprised community leaders, law enforcement officers, and health workers.


*[…] we have another support service within the community group. Which is actually a community group, it was founded by the Chamen’s foundation. It comprises of the police, health workers, community people like the Alfa, imams, and pastor they form a social community network that fight against abuse.*

*(Health provider, Sango Lagos)*


Insecurity in the communities also increases the experience SGBV, especially rape, in the area. The majority of the communities in the study locations are characterized by open markets (without gates), and young men who live in the street were openly engaged in drug abuse of various dimensions. In an informal conversation, a young girl reported that they dared not walk on the street any time after 7 p.m. in the area to avoid being raped. This led to the efforts of some girls to prepare themselves ahead by using contraceptives to protect themselves from unwanted pregnancy.


*[…] I have also seen a scenario in—which a girl came to us and she told us that she was still a virgin though but she wants to do family planning because she knows that she can be raped at any time and she does not want to be pregnant because she understands the environment that she lives in. Even her grandmother consented to it because I was like—so the grandmother gave us the consent that she is actually right. So environment too.*

*(Health provider, Powerline Lagos)*


The few community members also formed a society composed of stakeholders to address SGBV in the area. They ensure that the police, health care providers, and community leaders are involved to address cases of SGBV in their community. A health provider in the Lagos elaborated:


*[…] Like I said the other time, okay I have remembered “The Child Protection Committee” that has been set up within the community. We do not have it in every community but we have it in a few communities. Child Protection Association so it is like a link within the community to fight such things.*

*(Health Provider, Agege Lagos)*


### 3.2. Providers’ Responses and Challenges in Treating Gender-Based Violence Survivors

#### 3.2.1. Health Providers’ Responses and Involvement in AYW Health Decisions

##### Health Providers’ Attitudes to AYW and Prompt Intervention

The attitudes of health providers reportedly varied by the type of facility used by the survivors. Participants reported that in most secondary or tertiary hospitals, healthcare providers were willing to find out what was responsible for the pain and offer a solution or make referrals. This was unlike private hospitals where health workers did not care about what happened to the survivor; they would rather give treatments. Ironically, most survivors of violence who were more financially stable preferred to patronize these private hospitals. As emphasized by a participant in a FGD, she preferred the anonymity and less personal treatment. Treatment in the context of the private hospital was considered preferable by survivors of violence, especially AYW who would not need to answer any interrogation from health providers. This may be a result of expecting the story to remain in the treatment room. Hence, the decision of some survivors was to patronize private hospitals, as the health providers there were less concerned about the stories behind the abuse:


*[…] Adolescents that will go to hospitals go to private hospitals, they cannot go to general hospitals because they will counsel them there and they are not comfortable with it. General hospitals will want to get to the root of the matter, for example, maybe you go there and tell them your husband beat you and from there they will start probing wanting to know what is going on within the person’s family, unlike private hospitals. They are just after their money, just pay them and they will treat you without a story, they are less concerned about what is happening to you, just give them their money and you are good to go […].*

*(Resp. 3, FGD AYW Sango Lagos)*


Some health providers in few communities acknowledged their limitations in the area of SGBV and work with the state and NGO to refer cases of SGBV reported to appropriate channel that the survivors would be able to get adequate care and attention. This process was elaborated by a health provider in Lagos:


*[…] They come to us. We have contact with Lagos State where we can contact them and get in touch. We have some NGO that we work together like Hello Lagos. Hello, Lagos has their branch at the Lagos University Teaching hospital (LASUTH), also at Agege too. So when such a case is been found, it’s either I report to Hello Lagos LASUTH or I put to my office directly and they take it up, you know, ask the girl to come, fix her up. Then go through the protocol, then to Lagos State Domestic Violence Respond Squad […].*

*(Health provider, Powerline Lagos)*


##### Support by Health Providers for Survivors Who Utilized Services

In most cases, healthcare providers first offered counseling and then referred the young women to places where they could get treatment and care, and in cases of very severe violence, they were referred to places where they could actually report the case.


*‘[…] Like the one I referred to the other time the girl, seventeen years old was actually with her estranged boyfriend and she was stabbed with bottles in her hand and she had a baby and the baby was subtly taken away from her and all those stuffs. The mother came in and I had to refer the mother to the hmmmm- there is a center in the maternity that deals with violence […]’.*

*(Health Counsellor, Agege Lagos)*


One of the ways the health providers detected potential survivors was through observation of the patients who came to the clinic. According to one of the health providers, survivors often showed signs of depression and most times refused to socialize with other patients, which usually gave them the impression that there could be some disturbing personal issues the patient was hiding. Though observation is one of the guidelines from the WHO by which health providers could identify survivors or those currently in abusive relationships, unfortunately, only few used this approach. This importance of being sensitive and observing patients becomes evident in the following elaboration:


*‘[…] So whenever she comes for the clinic, she does not interact, laugh or smile. Then one day, I called her and asked her what the problem was, it was there she narrated the story that her mother-in-law was not treating her well and her parents had warned her never to come back home. The University College Hospital (UCH) organized one program/project on adolescents’ pregnancy. I had to inform them; they went to the place and settled the matter because I noticed she was already becoming depressed […]’.*

*(Matron Ayeye, Ibadan)*


In PHC, the providers are still limited in providing adequate support to the survivor, as the issue of referral keeps reoccurring in the facilities, which limits the support health providers could offer on the spot:


*‘[…] Unless only if the survivor is injured, we take it up just to save the life of the girl, because sometimes if we refer them to ‘Adeoyo’ hospital, they will not go. So, we try and do some tests for them, like HIV test and give them and if there is none, we move further to a pregnancy test, that is our limit, we don’t have any power other than that.*

*(Matron Oniyanrin Ibadan)*


##### Health Providers’ Challenges in Supporting Violence Survivors

Healthcare providers were concerned about the limited support they could provide to survivors, as a result of the shortage of available tools and professionals which the government provided in primary healthcare centres in the community.


*[…] There are no social workers in this community, if there were, they would be confronted with such problems as the woman who said she wanted to commit suicide […].*

*(Matron Ayeye Ibadan)*


Prominent among these limitations were the absence of social workers to address some particular needs of survivors, such as counselling. In some instances, the need to cater for the aggressors too, in terms of rehabilitation and reformation, as presently, there is more emphasis on punishment of the aggressors via imprisonment, which could be counter-productive:


*‘[…] In every local government, I believe there is a need to have a rehabilitation center for abuse or cases of abuse. Like now, I have case of abuse and I am going to Ikeja general hospital […]’. ‘[…] the survivors of GBV you can counsel them like I said but the treatment would not be entirely 100% fixed. The treatment would not be that comprehensive within the primary health center within the locality because we do not have a particular unit of it, so, if such an office is responsible for just gender-based violence for young people, it would actually go a long way […]’.*

*(Health providers, Agege Lagos)*


One of the concerns of health providers was the process at which perpetrators are prosecuted and released back to the society. One of the participants mentioned rehabilitating the perpetrators by having centers within each local governments in the state that focus on this aspect:


*[…] most of these offenders we do not take them to rehabilitation homes before they are jailed in prison or releasing them back into the community. They need to undergo rehabilitation; we need to have rehabilitation centers probably within the local government at least two or three centers within the local government.*

*(Health counselor, Agege Lagos)*


The challenges faced by health care providers in providing adequate support were summarized in Figure 2 below: lack of psychosocial support, distance to support center, referral problems, lack of social workers, lack of equipment, lack of rehabilitation center for survivors of GBV and lack of gender-based violence office in community PHCs.

See Figure 2.

**Figure 2 healthcare-11-02627-f002:**
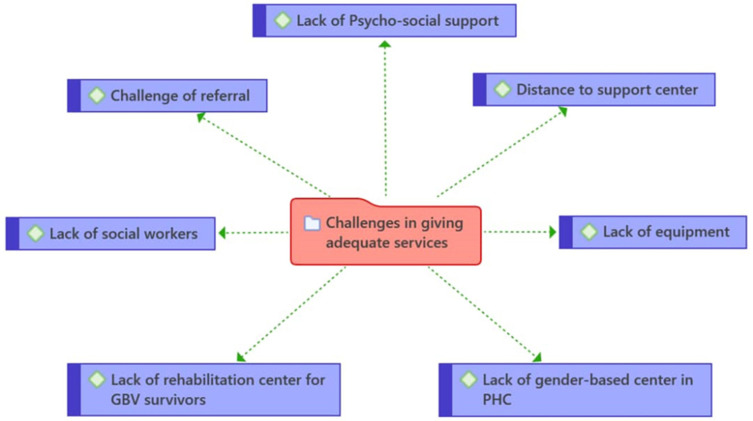
Shows challenges in providing adequate services by providers in PHC.

##### Health Providers’ Professionalism in Treating Survivors of GBV

Health providers need to observe and monitor patients who have been victimized in order to promptly provide the necessary support. It was evident that there were no professional health providers such as psychiatrists, psychologists, and counselors in the study sites to assist abused AYW. Incidentally, the primary healthcare facilities are closer to community members. One of the provider mentioned a complete service provision for the survivor very necessary at the PHC:


*[…] The major reason why they need to go through this support service is to get better services. Better services in terms of psycho-social support services, better services on health, and better services in managing the survivors and the perpetrators also. Better services to manage the girl child to be able to fit in back into the community […].*

*(Health provider, Agege Lagos)*


Most of the health providers admitted the limitation of help they could render to the survivors. One of the providers mentioned that they do not have all the help urged the survivors to voice out to get further help:


*[…] Like I told you, we don’t have much to meet, we don’t have a solution, we just counsel the person to voice out to look for the solution, like me now. I only advised them to please voice out to look for a solution […].*

*(Matron Ayeye Ibadan)*


Figure 1 shows the visualization of perceived barriers to treatment encountered by AYW and the variation by location. Although the two study sites are in urban low-income communities, the culture of silence and other social norms, and individual perspectives of the survivors limit their possibilities in accessing treatment for SGBV. The thickness of the grid shows the magnitude and frequency level of participants showing barriers to accessing health services after GBV experience. For example, more AYW in Ibadan than in Lagos had difficulties accessing healthcare facilities due to stigma from friends, neighbors, and possibly health providers. Lack of financial resources was another barrier in accessing the health system, which was emphasized more often in Ibadan compared to Lagos; it could be as a result of subsidized services. One of the significant barriers mentioned that was at the same frequency in both locations was the issue of delay and access to justice. In comparison, shyness and fear of stigmatization were far higher in Ibadan compared to Lagos; threats by partners were the same in both sites as barriers to health service use, and lack of funds was higher in Ibadan compared to Lagos. Societal acceptance and children were higher in Ibadan compared to Lagos as barriers. In all, fear of stigmatization, lack of funds, delay in justice, societal acceptance, having children, and threat were the major barriers in accessing health services by the participants. All of the barriers identified in Figure 1 were different in the two locations.

## 4. Discussion

This study set out to understand barriers faced by AYW in accessing health facilities for treatment on GBV, and the perspective of health providers on access, previous support rendered by providers, and challenges in giving needed support to survivors. The findings in this study showed a range of interconnected factors operating at individual community and institutional or health system levels that influenced AYW decisions of choices on health facilities that meet their needs. The culture of silence is one of the reasons mentioned as a barrier of not reporting, as speaking out about sexual status may bring about shame, stigma, being-judged, blamed, lack of support from the community members, and fear of re-victimization from the perpetrators. This is consistent with previous studies on the decisions of survivors of IPV either to keep silent about their experience or to run away from the relationship [17,18]. The findings affirm the extent of interactions, which are determined by the social/economic status present as a pathway of vulnerability or privilege among AYW (individuals) and institutions and structures (health providers/health facilities). The relatively young age of AYW put them in a disadvantaged group that limited the extent of access to support services, which may not be the case for older women. For instance, AYW were seen as children who should not have a partner or experience IPV in the first place while this perception limits their abilities of approaching health facilities for support. As the results showed, they would rather prefer to go to a pharmacy store or private clinic for treatment, which offers limited services.

From the providers’ perspectives, it was emphasized that there were no special clinics for young people experiencing SGBV unlike YF-SRHS in PHC facilities in the community, i.e., care for sexual well-being of young people, such as contraception, family planning service, HIV counseling and testing, STI screening and treatment, and others. The confidence of AYW in approaching the health providers and the likelihood of expectations to receive care were limited. This result corroborated a study from Switzerland on migrants who experienced GBV [20], where their lacking confidence in approaching healthcare systems was described as a major barrier to the disclosure of GBV. In addition, the participants of our study belong to rather marginalized groups due to age, socio-economic status, and place of living. Incorporating GBV services in the form of YF-SRHS clinics could be one of the approaches to help AYW voice their experiences of SGBV in their intimate relationships, as this will give free access and trust to the health system and assure confidentiality. It also gives reassurance of access to health professionals, such as psychiatrists, psychologists, and counselors, for recuperating the survivors during a longer process. A study on the implementation of family planning clinic-based partner violence supported that the system-level barrier confronted by providers limited the extent of support they could offer to victims/survivors of IPV [21].

The above-mentioned disadvantages in providing support to victims of violence are more pronounced in PHC facilities, as most of the cases were being referred to state hospitals for proper treatment and further interventions. It was only one of the communities of this study that had a SGBV office, which is in the PHC at Sango Agege LGA. Even with that, the majority of cases of SGBV were being referred to Lagos State University Teaching Hospital (LASUTH). Most of the AYW did not go to LASUTH nor the University College Hospital (UCH) when referred due to the distance and complexity of the referral structure. It was evident that the providers were displeased with the process and their lacking possibilities to give adequate support to survivors, and the concern for limited reports was due to process-related challenges. A study by Garcia and others [25] seems to be divergent according to the findings, as most of the primary healthcare centers in the study communities were not well structured to offer SGBV services in such a way that will make AYW disclose experiences of violence in an intimate relationship, as these were not within their reach. Of course, this is compounded in low-income communities as those in the city were likely to enjoy state facilities compared to those in the more disadvantaged regions of the same city like the study areas. The role of health providers cannot be overemphasized in mitigating the experiences of IPV among AYW especially in the informal settings. Nonetheless, the WHO Manual on Health Systems Response [24] on IPV does not categorize facility locations; therefore, this needs a second look and political will from the state party to provide the availability of access to health professionals especially at the grassroots level, where many vulnerable AYW reside.

In this study, the use of the intersectionality lens allows us to take into account the various layers of inequality when it comes to AYW’s access to adequate services after having experienced violence and how these shape poor health outcomes in the more marginalized communities. As could be shown in the results section, their age, gender, and socio-economic position amongst others played an important role. The homogeneity of experiences is prevented due to locations (in terms of availability, accessibility, affordability, adequacy, and acceptability) [40,41], as these may not be the same as those in the urban center, close to tertiary health institutions. The study revealed multiple communities and structures of power that interacted in health systems, which produces multiple levels of disadvantage or privilege among AYW. It exposed the link between primary and tertiary healthcare services depending on the locations, the multifaceted barriers due to access-(shaped by age, social norms, value, community perceptions, and health systems structures) and political will for the availability of health professionals’ services on SGBV at the local government level that will mitigate health inequalities [42]. The significance of this study, first, is that it reveals that the low reports of experiences of SGBV by survivors are due to a range of interrelated factors, including the lack of awareness of possible health seeking pathways, the financial situation of survivors, stigmatization, family perspective to SGBV (including the “culture of silence”), and inadequate availability of support from the PHC system in the community in terms of available professionals (psychiatrists, psychologists, counselors) beyond doctors’ and nurses’ diagnostic treatments. These factors are further complicated by the issue of concealing violence experiences as acceptable, and not revealing the SGBV status prevents survivors from reporting SGBV experiences. This is due to low awareness of the SGBV impact and very little evidence of justice, which makes perpetrators walk freely after the incident of violence in the community. Since the trauma of violence could be indelible in the memories of survivors, routine counselling and professional support for the survivors along this lengthy journey will go a long way to avoid re-victimization.

### 4.1. Strengths and Limitations

The strengths of this study were to understand how survivors of IPV encountered barriers in accessing support from the health system and the available support received. The institutional structure and cultural barriers encountered in accessing the facilities, the level of reporting and adequate care received from health professionals were not sufficiently available in the study locations. This study also revealed the need for specialized health professionals, such as psychiatrists, psychologists, and counselors, next to doctors’ and nurses’ treatments in order to enhance full- and long-term support for the survivors. However, some limitations need to be discussed. First, our findings are based on self-reports from the survivors and health providers, and we cannot control the level of bias as AYW with negative experiences of IPV may have great challenges to share or discuss them in general but also with the study team. However, after spending a considerable amount of time in the community, relationships were built, and participants started to open up to the research team. These were encountered during our data collection as well as the emotional trauma of bringing back old, painful memories of SGBV. Second, the data collection process was a rapid approach due to the outbreak of COVID-19 pandemic at the period. The challenge of gaining trust from the participants was difficult at first, as they initially thought that the first author and research assistants were government officials involved in the COVID-19 treatment and immunization. In addition, our study was not able to interview other health professionals apart from doctors and nurses, as the other professionals were not available in the facilities within the study locations. We also did not ask the participants about their needs for health professionals in the process of recovery or if a lack of health professionals was a barrier to utilizing health services. Our results should be interpreted with care, as this is only applicable to the locations of the study and should not be generalized. The study shows a pathway in which AYW could utilize health facilities, and the provision of all the necessary arms of the health system available in primary health centers.

### 4.2. Implications for Policy

Given the high prevalence of cases, health providers need retraining on how to handle AYW who are survivors of abuse and SGBV violence, as prompt interventions are required by providers to mitigate and avoid falling back into harmful relationships. The Ministry of Health in Nigeria is in a position to channel the retraining of health providers in the state towards the course of mitigating SGBV, but this should be adapted to local context as recommended by the WHO guidelines. In addition, psychologists, psychiatrists, periodic counseling for survivors, and rehabilitation centers for survivors as well as for perpetrators were not well positioned to provide support to victims of violence in the health system. A support system for SGBV survivors is either inadequately developed or was not available in the primary healthcare centers we investigated. While conducting an intensive study across all of Nigeria is infeasible, we believe that the overarching findings are transferable [43] to other settings within the country as well. Hence, the need for appropriate intervention to make the health system accessible and available for support to mitigate the consequences of violence among female AYW.

However, if necessary structures are not in place, this may be difficult to achieve. The political will is important, especially when it comes to these marginalized urban areas to make health system responses to violence against women and girls achievable. Therefore, the federal government should pay more attention to the implementation and evaluation of what works best in the fight against GBV. The involvement of stakeholders in advocating for the eradication of GBV can be achieved through leadership in the health sectors, public reports of violence, media coverage, civil society advocacy, and enlightening leaders about the magnitude of IPV and its link to poor health outcomes. These elements are needed for improving the health system’s involvement and commitment [24].

## 5. Conclusions

As demonstrated by this study, AYW in low-income communities in urban areas are particularly vulnerable to the long-term effects of SGBV. As illustrated above, a culture of silence and the lack of adequate social policies impact their decisions and pathways of seeking for support. Further, a greater commitment channeling of resources, setting priorities in order to invest in needed trainings and trusted grassroots facilities, and a broader network of providers who can support the survivors of SGBV. However, those in intimate relationships experiencing violence are the consequence of policy priorities and decisions that can be changed. Health systems can take over the role of safeguarding survivors of violence and stand up for their rights, as some survivors themselves do not have the power to take someone to court due to the various intersecting layers of inequality mentioned above. Education and empowerment of AYW are fundamental in the disclosure of SGBV. Health providers should be trained to identify and support survivors and strategies to address SGBV. In addition, integrating SGBV services into primary health care to target AYW, such as YF-SRHS, is a way forward to reduce SGBV experience or repeat victimization among young people. Decentralization of gender-based units at the PHC level gives AYW confidence to report and get the necessary care for GBV promptly and without further cost. Strengthening the young women through their agency, such as society and groups, will give support beyond the individual levels. Beyond a reactionary, curative approach, health providers need to play a preventive role through education, advocacy, and coordination of interventions at the community level.

## Figures and Tables

**Figure 1 healthcare-11-02627-f001:**
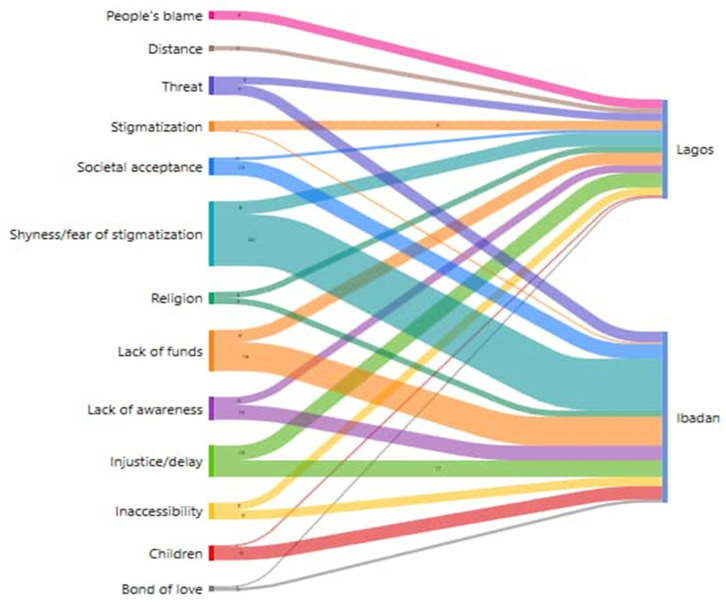
Shows various barriers by AYW in reporting and to accessing health services (Barrier in reporting and accessing health facilities for treatment of GBV).

## Data Availability

The dataset presented in this article are not publicly available, because it contains information that could compromise the privacy of the interviewees and a breach of agreement. Request to access the dataset can be directed to the corresponding author. Author details: Department of Epidemiology and Public Health, Swiss Tropical and Public Health Institute, Basel, Switzerland, University of Basel, Switzerland, Obafemi Awolowo University Ile Ife, Nigeria^,^ Mixed Method Program and Faculty of Medicine, University of Michigan, United States.

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
