# Peer review of "Health Providers’ Response to Female Adolescent Survivors of Sexual and Gender-Based Violence and Demand Side Barriers in the Utilization of Support Services in Urban Low-Income Communities of Nigeria"

_healthcare, 2023, doi:10.3390/healthcare11192627_

Round 1
Reviewer 1 Report
Dear authors, the manuscript needs major improvements in order to be published. The references used are very outdated, they should be of recent publication, from the last 5 years, furthermore, the qualitative study has major shortcomings, it lacks validity criteria such as credibility, saturation, transferability and confirmability. It is recommended to read Ruiz Olibuenága.
The article in general does not follow the format of the journal, especially the bibliographical references.
In general, the article lacks sufficient and verifiable quality to be published in a journal of the scope of Healthcare, so it is recommended that it be rewritten in accordance with the current state of the scientific field in the subject (last 5 years) and that interviews be carried out with appropriate methodology and with a more complete discussion, as well as limitations and proposals for future lines of research.
Author Response
Dear Reviewer,
Reviewer 1 comments and response
Reviewer comments |
Responses |
Dear authors, the manuscript needs major improvements in order to be published. |
We have taken time to improve on the manuscript |
The references used are very outdated, they should be of recent publication, from the last 5 years |
The references have been updated. However, other references earlier than 5 years are very important. |
The qualitative study has major shortcomings, it lacks validity criteria such as credibility, saturation, transferability, and confirmability |
The study protocol is explicitly stated. The audio recordings of interviews, transcripts, coding, and further analysis are available on request. More details have been provided in the revised manuscript of procedures of data collection as well as the description of the study locations. The cultural settings of the study locations have also been explained further to enable transferability. |
The article in general does not follow the format of the journal, especially the bibliographical references |
This has been corrected to follow the journal format, including the bibliography |
In general, the article lacks sufficient and verifiable quality to be published in a journal of the scope of Healthcare. |
One of the verifiable qualities is that, it was ethically approved by the Ethics Committee, Northwest and Central Switzerland and the University of Ibadan Research Ethics Committee. The study was carried out in the specified locations and the subject matter is a current concern in healthcare service |
So it is recommended that it be rewritten in accordance with the current state of the scientific field in the subject (last 5 years |
Updated current references have been included. Those earlier than five years are also considered very important references. |
Interviews be carried out with appropriate methodology and with a more complete discussion, as well as limitations and proposals for future lines of research. |
The current manuscript is extracted from a larger ethnographic study, which is why the overall methodology is still retained. However, we have mentioned this as part of the study limitations. |
For more details, please see the revised manuscript.
Reviewer 2 Report
Thank you for this interesting paper, which I enjoyed reading. It reports on an important study exploring the nature of the support system for young women and female adolescents’ following sexual violence.
The paper is well written and structured overall. I have some minor comments that I hope are helpful. It also requires a final proofread to address minor typos.
One general query – what’s the difference between a ‘victim’ and ‘survivor’ (in the text often indicated as ‘victim/survivor’)? I would tend towards using ‘survivor’ (‘victim’ can be a degrading term, implying passivity). I’d recommend you think further about this. It may depend on whether or not the person is still experiencing violence.
Also, GBV and IPV are used at different times in the text. They’re not necessarily interchangeable (though will overlap) but have specific definitions. I’d recommend being very precise here.
BACKGROUND
This section provides a useful and detailed background to the topic, and justification of the study in Nigeria. Taking an approach based on intersectionality is valid.
METHODS
This provides details of how the sample were recruited and data collected and analysed. It is indeed taking a qualitative approach, though whether or not it’s ethnographic is a different question. What makes this ethnographic as compared to, say, a purely qualitative study using interviews and focus groups? This isn’t a serious problem as such (it doesn’t affect the quality of the study) but, given the very specific definition of ethnography (which usually involves some form of observational data alongside data gathered from interviews and focus groups, etc.) and is very focused on cultural constructs and beliefs, you need to make clearer what the ethnographic elements actually are.
In section 2.3, you don’t specify the recruitment procedure. Can you provide more details here?
In section 2.5, given the very sensitive nature of the research topic, I would like to see more here abouts the safety of respondents and how they were protected and supported during and after the interviews and focus groups, especially survivors of abuse and violence.
The analysis to generate the sub-categories seems appropriate, and I note that ethical approval was granted.
RESULTS
This is a strong section and includes some important insights into the topic from the perspective of people with experience of sexual violence, and their carers. I was interested especially in the barriers to accessing care, and the impact of stigma.
The use of interview excerpts is illuminating but check you don’t replicate your examples. There is at least one example used twice, on line 315 and again 416). I would tend to use excerpts sparingly.
I like the diagram (figure 1) – I’m interested to know how you defined the thickness of each line. Was this based on a quantitative word count on specific terms appearing in interviews and focus groups? Or is it more arbitrary? It would be helpful for readers to know.
DISCUSSION AND CONCLUSION
You relate your findings to what is currently known in the field, and this is helpful. You include suitable citations and references to other studies but also highlight the contribution your study has made. Your limitations are appropriate and valid, and you make reasonable recommendations for policy.
The conclusion provides a useful summary.
REVIEWER RECOMMENDATIONS
1. Check use of terminology (see comments), especially around victim/survivor, ethnography, and GBV and IPV.
2. Add more detail about the recruitment procedure.
3. Add more detail about safety and protection of respondents.
4. Review interview excerpts to exclude duplications.
5. Add a note to explain how the thickness of lines for figure 1 were decided.
The quality of the English language is high, but there are some minor aberrations that need correction.
Author Response
Dear Reviewer,
Reviewer 2 comments and response
Reviewer’s comments |
Responses |
One general query – What’s the difference between a ‘victim’ and ‘survivor’ (in the text often indicated as ‘victim/survivor’)? I would tend towards using ‘survivor’ (‘victim’ can be a degrading term, implying passivity). I’d recommend you think further about this. It may depend on whether or not the person is still experiencing violence. |
Thank you for your response. We have retained the concept of survivor all through the article. |
Given the very specific definition of ethnography (which usually involves some form of observational data alongside data gathered from interviews and focus groups, etc.) and is very focused on cultural constructs and beliefs, you need to make clearer what the ethnographic elements actually are. |
This has been duly attended to in the manuscript ( see the methods section) |
In section 2.3, you don’t specify the recruitment procedure. Can you provide more details here? |
This has been clarified |
In section 2.5, given the very sensitive nature of the research topic, I would like to see more here about the safety of respondents and how they were protected and supported during and after the interviews and focus groups, especially survivors of abuse and violence. |
The authors ensured that information collected from the survivors were kept confidential and all reports were written anonymously to ensure that participants are protected from any reprisals. The process of data collection also ensured that privacy of the participants were maintained as one on one interviews were held with survivors with no interference. |
The use of interview excerpts is illuminating but check you don’t replicate your examples. There is at least one example used twice, on line 315 and again 416). I would tend to use excerpts sparingly. |
This has been duly noted and corrected |
I like the diagram (figure 1) – I’m interested to know how you defined the thickness of each line. Was this based on a quantitative word count on specific terms appearing in interviews and focus groups? Or is it more arbitrary? It would be helpful for readers to know. |
Figure 1 shows the magnitude & frequency levels of participants showing barriers to utilizing health services. |
REVIEWER RECOMMENDATIONS 1. Check the use of terminology (see comments), especially around victim/survivor, ethnography, GBV, and IPV. |
This has been addressed in the updated manuscript |
2. Add more detail about the recruitment procedure |
More detail has been added to section 2.3 |
3. Add more detail about the safety and protection of respondents. |
See comments above |
4. Review interview excerpts to exclude duplications. |
See comments above |
5. Add a note to explain how the thickness of the lines for Figure 1 were decided. |
See comments above |
For more details, please see the revised manuscript.
Reviewer 3 Report
This article incorporates an ethnographic approach to assess health providers’ response to female adolescents’ survivors of sexual violence, and demand side barriers in the utilization of support services in urban shantytowns of Nigeria. Authors contend that survivors of sexual and gender-based violence (SGBV) often face barriers when trying to access quality healthcare which encourages a need to understand the context of health providers’ inadequacy and structural issues of health services in providing comprehensive and inclusive support services to victims of SGBV to meet their needs. The review is as follows.
1. Within the Background, explain the meaning of ‘demand side’ and expand discission on its relevance to the paper topic.
2. Line 58-59 – Check wording in “A study from Calabar among teenage girls found that beating by sexual partners and guardians were prevalent in the study area [11]…”. Consider make the word plural (beatings).
3. In the Background, there is an important mention of the importance of political will in making health system responses to violence against women and girls achievable. (lines 105-106)
4. Line 161 – Check for an extra space after the word ‘discussions’.
5. Lines 161-162 – In “…were used in eliciting information from 10 health providers, 40 adolescent’ girls and young women”, is the number 40 referring to adolescents’ girls and young women in total? If so, consider saying, “10 health providers and 40 adolescent girls and young women”. Also, remove the apostrophe in ‘adolescents’.
6. For the Methods and the 45-minute survey, were incentives given to interviewees to participate in the study?
7. In Figure 1 (page 10), it is difficult to decipher the chart. Maybe a guide can be included to explain what the different colors mean. Also, it appears there are numbers within the lines. If so, they are hard to read. Consider increasing the font of the numbers.
8. For implications for policy, expand discussion on the need for retraining on how to handle AYW (line 533). Who will deliver the training and what will the training entail? Consider discussion on societal norms and political will as additional factors that can influence policy development to address sexual and gender-based violence towards female adolescents.
Overall, this is an insightful, relevant paper on a compelling topic. It would help address a gap in the existing literature. Attending to the suggested feedback can help make the paper clearer, more comprehensive, and ultimately suitable for publication.
Consider a proofread for English language and style.
Author Response
Dear Reviewer,
Reviewer 3 comments and responses
Reviewer’s comments |
Responses |
1. Within the Background, explain the meaning of ‘demand side’ and expand the discussion on its relevance to the paper topic.
|
Thank you. This has been explained better |
2. Line 58-59 – Check the wording in “A study from Calabar among teenage girls found that beating by sexual partners and guardians were prevalent in the study area [11]…”. Consider making the word plural (beatings).
|
Thank you. This has been corrected |
3. In the Background, there is an important mention of the importance of political will in making health system responses to violence against women and girls achievable. (lines 105-106) |
Thank you. The sentence has been elaborated. |
4. Line 161 – Check for an extra space after the word ‘discussions’. |
This has been corrected |
5. Lines 161-162 – In “…were used in eliciting information from 10 health providers, 40 adolescent’ girls, and young women”, is the number 40 referring to adolescents’ girls and young women in total? If so, consider saying, “10 health providers and 40 adolescent girls and young women”. Also, remove the apostrophe in ‘adolescents’. |
Thank you for the correction. This has been corrected. |
6. For the Methods and the 45-minute survey, were incentives given to interviewees to participate in the study? |
Yes! Transportation was arranged for FGD participants to enable them get to the place of interview which was agreed by all. Toiletries were given to girls and a moderate amount of call cards for health providers |
7. In Figure 1 (page 10), it is difficult to decipher the chart. Maybe a guide can be included to explain what the different colors mean. Also, it appears there are numbers within the lines. If so, they are hard to read. Consider increasing the font of the numbers. |
The explanation of the figure has been clearly stated. The color only specifies different barriers encountered. |
8. For implications for policy, expand the discussion on the need for retraining on how to handle AYW (line 533). Who will deliver the training and what will the training entail? Consider discussion on societal norms and political will as additional factors that can influence policy development to address sexual and gender-based violence toward female adolescents. |
Thank you. This has been considerably attended to in the manuscript. |
For more details, please see the revised manuscript.
Reviewer 4 Report
The article titled "Health Providers’ Response to Female Adolescents’ Survivors of Sexual Violence, and Demand Side Barriers in The Utilization of Support Services in Urban Slums of Nigeria " presents a comprehensive exploration of the challenges faced by young girls and women in Nigeria concerning gender-based violence (GBV) and the role of the healthcare system in providing support. The article effectively sheds light on a critical issue, offering insights that can contribute to positive changes in policies and interventions. Here are some positive points about the article:
Comprehensive Background: The article provides a well-detailed background that highlights the global issue of gender-based violence faced by young girls and women, backed by relevant statistics. This introduction effectively draws attention to the urgency of addressing the issue and its impact on the well-being of adolescents.
Global Relevance: By situating the issue within the context of Sustainable Development Goals and the United Nations' recognition of the significance of adolescents in achieving broader societal goals, the article shows the global relevance of the issue and the need for concerted efforts.
Thorough Literature Review: The article effectively integrates existing literature, showcasing the depth of research conducted on gender-based violence and related matters. This gives credibility to the article and demonstrates the authors' thorough understanding of the topic.
Appropriate Methodology: The study's methodology is robust, utilizing a combination of qualitative research methods to gather data from various perspectives, including both survivors of violence and healthcare providers. This approach enhances the credibility of the findings and enriches the overall analysis.
Intersectionality Perspective: The article's inclusion of the intersectionality theory to analyze the experiences of oppression and marginalization adds depth to the study. This perspective acknowledges the complexity of different social identities that influence individuals' experiences and vulnerability to violence.
Qualitative Analysis: The article employs a rigorous approach to qualitative data analysis, using established techniques like thematic content analysis and framework analysis. This ensures that the findings are derived from a systematic and comprehensive examination of the data.
Overall, the article makes a valuable contribution to the existing body of knowledge, offering insights that can lead to positive changes in addressing and preventing gender-based violence and improving healthcare support for survivors.
***Areas for improvement
While the article is commendable in many aspects, there are a few areas where further improvement could enhance its overall impact and clarity:
1-Clear Objectives: While the article briefly mentions the objectives of the study, it could benefit from a more explicit statement of its research objectives at the beginning. Clearly outlining what the study aims to achieve would provide readers with a roadmap for understanding the study's focus.
2-Contextual Background: Although the article provides a comprehensive global background on gender-based violence, it might be helpful to include more context-specific information about Nigeria. This could include socio-cultural factors, legal frameworks, and other contextual elements that shape the issue within the country.
3-Participants: We need a table describing the sociodemographic and related health characteristics of each participant, along with their abuse type and any other important highlights. Such a table could enhance the readers' understanding of the participants' backgrounds and experiences, allowing for more nuanced interpretations of the study's findings. This addition would help establish context and facilitate comparisons across participants.
Sample table:
Participant ID |
Age |
Marital status |
Education |
Employment |
Location |
Abuse type |
|
|
|
|
|
|
|
4-Figure one needs a clear legend
Thank you
Author Response
Dear Reviewer,
Reviewer 4 comments and responses
Reviewer’s comments |
Responses |
1-Clear Objectives: While the article briefly mentions the objectives of the study, it could benefit from a more explicit statement of its research objectives at the beginning. Clearly outlining what the study aims to achieve would provide readers with a roadmap for understanding the study's focus. |
Thank you for the comment. Please see the background. The research objectives have been well stated. |
2-Contextual Background: Although the article provides a comprehensive global background on gender-based violence, it might be helpful to include more context-specific information about Nigeria. This could include socio-cultural factors, legal frameworks, and other contextual elements that shape the issue within the country. |
Thank you. The socio-cultural context of Nigeria and the legal framework of sexual and gender-based violence have been added to the background |
3-Participants: We need a table describing the sociodemographic and related health characteristics of each participant, along with their abuse type and any other important highlights. Such a table could enhance the readers' understanding of the participants' backgrounds and experiences, allowing for more nuanced interpretations of the study's findings. This addition would help establish context and facilitate comparisons across participants. |
Thank you. This has been stated in descriptive form. |
Sample table:
|
The sociodemographic table has been adapted in written form to avoid looking quantitative. |
4-Figure one needs a clear legend |
Figure one has been duly explained |
For more details, please see the revised manuscript.
Round 2
Reviewer 1 Report
Revisions have been carried out properly
Author Response
A colleague who is fluent in English thoroughly revised the manuscript.
All refences were checked for their relevance to the manuscript.
Reviewer 3 Report
The revised version of the paper is clearer and more comprehensive. A few items to note:
1. Good clarification of the word ‘demand side’. It seems that ‘demand side’ is another way to say ‘need for services’.
2. Lines 165-167 – Check comma placement in the sentences ‘We conducted KII with 10 health providers 40 IDI with adolescent girls, and young women and nine sessions of focus group discussions with a 166 minimum of 8 participants in each group from January 2021 to June 2021 in both locations’. Consider rewording the sentence to: ‘We conducted KII with 10 health providers, IDI with 40 adolescent girls and young women, and nine sessions of focus group discussions with a 166 minimum of 8 participants in each group from January 2021 to June 2021 in both locations’.
3. There is a good mention of incentives (e.g., toiletries, call cards) provided to participants in the author feedback. This is good information that can be added to the paper.
4. Good explanation for the chart (figure 1). Also explain if there is a gradient to the barriers (e.g., Do AYW experience more or less of certain barriers?).
5. Lines 734-735 – Check the underlining in the words ‘primary health services to target AYW’.
Overall, the paper is improved. Attending to some remaining items should strengthen the paper and make it suitable for publication.
The revised version of the paper is clearer and more comprehensive. A few items to note:
1. Good clarification of the word ‘demand side’. It seems that ‘demand side’ is another way to say ‘need for services’.
2. Lines 165-167 – Check comma placement in the sentences ‘We conducted KII with 10 health providers 40 IDI with adolescent girls, and young women and nine sessions of focus group discussions with a 166 minimum of 8 participants in each group from January 2021 to June 2021 in both locations’. Consider rewording the sentence to: ‘We conducted KII with 10 health providers, IDI with 40 adolescent girls and young women, and nine sessions of focus group discussions with a 166 minimum of 8 participants in each group from January 2021 to June 2021 in both locations’.
3. There is a good mention of incentives (e.g., toiletries, call cards) provided to participants in the author feedback. This is good information that can be added to the paper.
4. Good explanation for the chart (figure 1). Also explain if there is a gradient to the barriers (e.g., Do AYW experience more or less of certain barriers?).
5. Lines 734-735 – Check the underlining in the words ‘primary health services to target AYW’.
Overall, the paper is improved. Attending to some remaining items should strengthen the paper and make it suitable for publication.
Author Response
All comments were attended to in the manuscript.